# *Clostridioides difficile* Flagellin Activates the Intracellular NLRC4 Inflammasome

**DOI:** 10.3390/ijms232012366

**Published:** 2022-10-15

**Authors:** Hiba Chebly, Jean-Christophe Marvaud, Layale Safa, Assem Khalil Elkak, Philippe Hussein Kobeissy, Imad Kansau, Cécile Larrazet

**Affiliations:** 1Institut Micalis, Université Paris-Saclay, INRAE, AgroParisTech, 91400 Orsay, France; 2Health Resources and Products Valorization Laboratory, Faculty of Pharmacy, Lebanese University, Beirut 1102-2801, Lebanon; 3Department of Natural Sciences, School of Arts and Sciences, Lebanese American University, Beirut 1102-2801, Lebanon

**Keywords:** *Clostridioides difficile*, flagellin, inflammasome, NLRC4, pro-caspase-1, gasdermin, IL-1β, IL-18, IL-33

## Abstract

*Clostridioides difficile* (*C. difficile*), is a major cause of nosocomial diarrhea and colitis. *C. difficile* flagellin FliC contributes toxins to gut inflammation by interacting with the immune Toll-like receptor 5 (TLR5) to activate nuclear factor-kappa B (NF-kB) and mitogen-activated protein kinase (MAPK) signaling pathways. Flagella of intracellular pathogens can activate the NLR family CARD domain-containing protein 4 (NLRC4) inflammasome pathway. In this study, we assessed whether flagellin of the extracellular bacterium *C. difficile* internalizes into epithelial cells and activates the NLRC4 inflammasome. Confocal microscopy showed internalization of recombinant green fluorescent protein (GFP)-FliC into intestinal Caco-2/TC7 cell line. Full-length GFP-FliC activates NLRC4 in Caco-2/TC7 cells in contrast to truncated GFP-FliC lacking the C-terminal region recognized by the inflammasome. FliC induced cleavage of pro-caspase-1 into two subunits, p20 and p10 as well as gasdermin D (GSDMD), suggesting the caspase-1 and NLRC4 inflammasome activation. In addition, colocalization of GFP-FliC and pro-caspase-1 was observed, indicating the FliC-dependent NLRC4 inflammasome activation. Overexpression of the inflammasome-related interleukin (interleukin (IL)-1β, IL-18, and IL-33) encoding genes as well as increasing of the IL-18 synthesis was detected after cell stimulation. Inhibition of I-kappa-B kinase alpha (IKK-α) decreased the FliC-dependent inflammasome interleukin gene expression suggesting a role of the NF-κB pathway in regulating inflammasome. Altogether, these results suggest that FliC internalizes into the Caco-2/TC7 cells and activates the intracellular NLRC4 inflammasome thus contributing to the inflammatory process of *C. difficile* infection.

## 1. Introduction

*Clostridioides difficile* is an anaerobic, spore-forming, Gram-positive bacterium found in both the environment and intestinal tracts of animals and humans [1]. *C. difficile* infection (CDI) is the leading cause of nosocomial antibiotic-associated diarrhea [2]. Since the past two decades, the emergence of highly virulent and epidemic *C. difficile* strains is associated with several international outbreaks [3]. The bacterium induces a strong intestinal inflammatory response that may lead to the development of pseudomembranous colitis as well as severe complications such as fulminant colitis, toxic megacolon, and colonic perforation. CDI is therefore associated with a high rate of morbi-mortality. The two large clostridial toxins, toxin A (TcdA) and toxin B (TcdB), have long been considered the main virulence factors of *C. difficile* [4] contributing to the inflammatory lesions observed in patients through the activation of major pro-inflammatory signaling pathways such as MAPK [5], NF-κB [6] and inflammasome [7].

Other *C. difficile* factors appear to be involved in the pathogenesis, such as the two S-layer proteins (SLP) [8], adhesins [9], hydrolytic enzymes [10], cell wall proteins [11], and flagella [12]. *C. difficile* flagellin FliC, the major component of flagella, plays an essential role in CDI-associated inflammatory response. FliC is recognized by cell surface-associated TLR5, one of the innate immune pattern recognition receptors (PRRs), leading to the activation of MAPK and NF-κB cell signaling pathways and secretion of pro-inflammatory cytokines (IL-8 and chemokine (C-C motif) ligand 20—CCL20) [12,13]. We previously showed that the presence of both, toxins and flagella is required for the development of mucosal lesions in a mouse model of CDI [14].

It is still unclear whether *C. difficile* flagellin activates other pro-inflammatory signaling pathways such as the inflammasome one. The inflammasomes play an essential role in connecting the detection of endogenous and microbial danger signals to caspase-1 activation and induction of protective immune responses. In intracellular pathogen bacteria, including *Legionella pneumophila* [15], *Shigella flexneri* [16], *Burkholderia thailandensis* [17], and *Salmonella* Typhimurium [18], flagellins present in the cytosol can also be recognized by NLR, nucleotide oligomerization domain receptors (NOD)-like receptors. Flagellin proteins interact with specific members of the neuronal apoptosis inhibitor protein (NAIP) family, which in turn drive assembly and subsequent activation of the NLRC4 inflammasome [19,20]. Activated inflammasomes recruit a bipartite protein known as apoptosis-associated speck-like protein containing a caspase activation and recruitment domain) (ASC) containing a caspase activation and recruitment domains, and drive the proteolytic activation of pro-caspase-1, to caspase-1, and GSDMD) activation, resulting in the release of mature, active IL-33, IL-18, and IL-1β [21] and membrane pores formation (pyroptosis) [22]. Therefore, activation of the inflammasome pathway by *C. difficile* flagella requires the internalization of FliC in the cytosol.

Previous studies showed that different flagellins from Gram-negative bacteria could also be internalized into epithelial cells via TLR5 [23]. We hypothesize that even though *C. difficile* is an extracellular bacterium, FliC could be internalized and activate inflammasomes, thus contributing to the inflammatory process and intestinal damages observed during CDI. In the present study, the internalization of *C. difficile* FliC inside the intestinal epithelial cells was assessed by confocal microscopy. We demonstrated for the first time that FliC is internalized and triggers NLRC4 inflammasome activation, leading to cleavage of pro-caspase-1 and release of inflammasome-related cytokines IL-18 and IL-33.

## 2. Results

### 2.1. C. difficile Flagellin Is Internalized by the Intestinal Epithelial Cells

Some bacterial factors such as toxins can penetrate the cells and alters their homeostasis. Internalization of *C. difficile* flagella has not yet been studied. To assay flagellin internalization, Caco-2/TC7 cells were incubated with *C. difficile* recombinant flagellin fused in N-terminal with the fluorescent protein GFP (GFP-FliC) or GFP, as a control, in a time-dependent manner. The C-terminal region of FliC, which is involved in inflammasome activation was kept intact. Internalization of *C. difficile* flagellin was assessed by a confocal microscope. The Mean Fluorescence Intensity (MFI) of GFP-FliC was monitored either at the optical z-sections at the apical pole of cells (“TC7-Surface”) or the optical middle z-sections corresponding to the inside of cells (“TC7-Intracellular”). Throughout incubation time fluorescence of the GFP control was not detected in any section. The GFP-FliC was mainly localized at the cell surface, following 2 h of incubation (Figure 1a,b). The intracellular MFI of GFP-FliC progressively increased before reaching its maximum at 6 h post-stimulation, while the cell surface signal presented an opposite pattern. (Figure 1b). The GFP-FliC was also observed at the optical middle z-sections at the respective incubation times (Figure 1a,b). These results show that FliC is internalized by the Caco-2/TC7 cells in a time-dependent manner.

### 2.2. C. difficile Flagellin Induces NLRC4 Activation

Flagella of intracellular pathogen induce inflammasome activation through phosphorylation of NLRC4 at Ser533. Internalization of *C. difficile* FliC into cells may therefore activate this same signaling pathway. NLRC4 activation was evaluated in Caco-2/TC7 cells stimulated with either *C. difficile* FliC, from *S.* Typhimurium (FlaST), known to activate NLRC4, or the NLRC4 inflammasome inducer *Bacillus anthracis* lethal factor (LFn)-Needle, as a positive control. The level of phosphorylated NLRC4 was then measured by western blot. As expected, LFn-Needle induced NLRC4 phosphorylation as short as 2 h following stimulation. Cells stimulation with FliC, or FlaST, induced the phosphorylation of NLRC4 in a time-dependent manner with a maximum level of phosphorylated protein observed after 6 h of incubation (Figure 2a,b). These observations were confirmed by using a truncated *C. difficile* flagellin (FliCΔ_286–290_) with five amino acids removed in the C-terminal part. The mutant protein lacks two leucine residues previously described as being essential for recognition by NAIP [24]. FliCΔ_286–290_ induced a lower NLRC4 phosphorylation level in stimulated Caco-2/TC7 cells compared to GFP-FliC and the two positive controls FlaST and LFn-Needle (Figure 2c). These observations indicate that *C. difficile* FliC activates NLRC4 via phosphorylation at Ser553, and this activation requires the presence of the C-terminal part of this flagellin.

### 2.3. C. difficile Flagellin Induces Caspase-1 and Gasdermin Activation

Following inflammasome formation and activation, caspase-1 is autoactivated and cleaved into two components p20 and p10. To evaluate whether FliC-mediated NLRC4 activation results in the activation of pro-caspase-1, cells were incubated with FliC or FlaST in a time-dependent manner (2 h, 4 h, 6 h, and 8 h), or with tumor necrosis factor alpha (TNFα) for 1 h as a positive control. The level of p20 protein was then measured by western blot. FliC or FlaST induced an increase in the release of the p20 subunit with a maximum at 4 h and 6 h respectively (Figure 3A,B). To confirm that p20 release is due to caspase-1 activation and subsequent cleavage, cells were treated with VX-756, a specific pro-caspase-1 inhibitor, at different concentrations for 4 h and then stimulated with FliC for 4 h, or with LFn-needle for 2 h, as a positive control. Releasing of p20 was significantly reduced in VX- 765-pretreated cells (Figure 3C,D). Similarly, the expression of the *CASP1* gene was increased in Caco-2/TC7 cells following stimulation with FliC (Appendix A). These results suggest that the release of p20 induced by *C. difficile* flagellin is mediated by the NLRC4 inflammasome and the subsequent activation of pro-caspase-1.

Activation of pro-inflammatory caspases can drive pyroptosis by the cleavage of GSDMD, resulting in the assembly of the cleaved GSDMD into the cytoplasmic membrane to form pores. GSDMD activation in the presence or not of the chemical inhibitor of caspase-1 VX-756 was evaluated. Cells were treated or not with different concentrations of VX-756 for 4 h and then stimulated with FliC for 4 h or with LFn-Needle for 2 h, as a positive control. The level of the non-activated form of GSDMD was then measured by western blot. Following FliC stimulation, a decrease in GSDMD level was observed, suggesting a cleavage of this protein (Figure 3E). Pretreatment of cells with VX-756 (at 25 and 50 µM), inhibited GSDMD cleavage following stimulation with either FliC or LFn-Needle (Figure 3F). Colocalization of GFP-FliC with the pro-caspase-1 inside the cells following 6 h of incubation was observed by confocal microscopy using a specific pro-caspase-1 antibody (Figure 3G). These data suggest that FliC induces the activation of pro-caspase-1 and the cleavage of GSDMD, via NLRC4 inflammasome activation.

### 2.4. C. difficile Flagellin Induces Inflammasome-Related Cytokines Gene Expression

Activation of NLRC4 inflammasome leads to specific cytokines synthesis such as IL-18, IL-33, and IL-1β. The expression of these cytokine genes in Caco-2/TC7 cells stimulated with FliC, FlaST, or FliCΔ_286–290_ was measured by real-time quantitative reverse transcription PCR (qRT-PCR) in a time-dependent manner. Upregulation of IL-18, IL-33, and IL-1β encoding genes was observed in FliC and FlaST -stimulated cells, compared to unstimulated cells (Figure 4). Levels of *IL18* (Figure 4a) and *IL33* gene transcripts (Figure 4b) reached a maximum after 6 h of stimulation with FliC (12-fold and 20-fold, respectively) whereas the expression of *IL1β* increased more rapidly and reached its highest level (20-fold) after 2 h of stimulation (Figure 4c). Similar gene expression profiles were observed after FlaST stimulation. Finally, FliCΔ_286–290_ did not induce the expression of the measured genes suggesting that the five amino acids deleted from the C-terminal region of FliC are essential for inflammasome-related cytokines gene expression (Figure 4a–c).

### 2.5. NF-κB Pathway Plays a Role in FliC-Induced Pro-Caspase-1 and Inflammasome-Related Cytokines Gene Expression

The pivot NF-κB pro-inflammatory signaling pathway plays a key role in inflammasome activity by priming the pro-interleukin and inflammasome effector expressions [25]. This regulation integrates proinflammatory signaling pathways such as those triggered by TLR ligation. NF-κB, however, prevents excessive inflammation and restrains NLR family CARD domain-containing protein 3 (NLRP3)-inflammasome activation through a poorly defined mechanism [26]. To evaluate the role of NF-κB in FliC-induced NLRC4 inflammasome activation, IΚΚα inhibitor BMS-345541 was used, allowing the inhibition of the NF-κB signaling pathway. Caco-2/TC7 cells were pretreated with the IΚΚα inhibitor (20 µM) for 1 h and stimulated with FliC in a time-dependent manner. The expression of pro-caspase-1, *IL18*, and *Il33* genes was then measured by qRT-PCR. As previously shown (Figure 4 and Appendix A), stimulation with FliC upregulated the expression of pro-caspase-1, IL-18, and IL-33 encoding genes. Inhibition of the NF-κB signaling completely abolished the expression of these genes (Figure 5). These results suggest that the NF-κB signaling pathway plays an essential role in the activation of the inflammasome signaling pathway.

### 2.6. C. difficile Flagellin Induces Inflammasome-Related IL-18 Synthesis

Inflammasome-related IL-18 synthesis in cell supernatants was assessed using an ELISA-based method. Caco-2/TC7 cells were stimulated with FliC, FlaST, or FliCΔ_286–290_ in a time-dependent manner. FliC, as well as FlaST, upregulated IL-18 synthesis following 6h of stimulation, compared to unstimulated cells (Figure 6). Following cell stimulation with FliCΔ_286–290_, an important decrease in IL-18 synthesis was observed, confirming the essential role of the C-terminal region of FliC in NLRC4 activation. Inhibition of the NF-κB signaling pathway using IΚΚα inhibitor (BMS-345541) abolished IL-18 synthesis in FliC- and FlaST-stimulated cells, suggesting that this pathway is involved in IL-18 synthesis following inflammasome activation. 

## 3. Discussion

Flagella from enteropathogenic bacteria are involved in the inflammatory host response which could be responsible for intestinal lesions. *C. difficile* is a pathogen known to induce severe inflammatory response involving toxins TcdA and TcdB which have been considered the major virulence factors. Nevertheless, a synergy between the toxins and the flagellin, the main component of the flagella, has been demonstrated in the induction of a strong deleterious innate immune response of the host [14]. The *C. difficile* toxins play a role in opening the tight junctions of the intestinal epithelium thus facilitating the access of the flagellin monomers to the innate immune receptor TLR5 expressed at the basolateral pole of the epithelial cells [27]. We previously reported that this FliC/TLR5 interaction activates the NF-κB and the MAPK signaling pathways which elicits the synthesis of pro-inflammatory cytokines which amplify the host inflammatory response and mucosal injury [12]. 

Flagellins from intracellular pathogens induce the formation of an inflammasome when present inside the cell and then trigger an inflammatory immune response which includes pyroptosis (a form of lytic programmed cell death) [28]. This inflammasome involves the NLRC4 protein characterized by its central NOD domain, a regulatory C-terminal leucine-rich repeat (LRR) domain, and an N-terminal part of variable structure, such as the caspase activation and recruitment domain (CARD) [29]. NAIP in humans (NAIP5 in mice) binds to the D0 domain of flagellins and assembles with NLRC4 to form the multiprotein inflammasome complex, which includes the adaptor protein ASC and the pro-caspase-1, via an interaction CARD-CARD [28]. The inflammasome serves as a platform to cleave pro-caspase-1 to caspase-1 which in turn induces the maturation of the pro-inflammatory IL-18, IL-33, and IL-1β cytokines, as well as the pore-forming GSDMD [30,31,32].

However, extracellular flagellin can be internalized by epithelial cells as observed for flagellins of *S.* Typhimurium and *Escherichia coli* strains which are internalized by polarized human Caco-2BBe and T-84 cells. This TLR5-mediated flagellin internalization follows the endosomal-dependent mechanism since flagellins co-localize with endosomal and lysosomal compartments [23]. Moreover, in plant cells, it has been described that the fragment flg22 of bacterial flagellin binds the Flagellin Sensitive 2 (FLS2) receptor and is internalized via the endocytic pathway to induce both local and systemic immune responses [33,34]. In the present study, confocal analyses with a tagged *C. difficile* flagellin demonstrate for the first time that flagellin from an extracellular Gram-positive pathogen can also be internalized by intestinal epithelial cells. The activation of NLRC4 inflammasome by the *C. difficile* FliC was expected since the *C. difficile* FliC shares with other flagellins some of the C-terminal essential amino acids recognized by NAIP5 (Appendix A) [24,35]. It has to be noticed that in FliC an arginine residue from C-terminus is replaced by glycine, as for the flagellins of the enteropathogen *E. coli* (EPEC) and enterohemorrhagic *E. coli* (EHEC), a configuration that seems to induce less NLRC4 activity [36]. Nevertheless, in this study, the internalization of the *C. difficile* FliC induced Ser533 phosphorylation of NLRC4 as previously described for intracellular pathogens. Interestingly, a significant decrease of NLRC4 Ser533 phosphorylation was observed with the truncated FliCΔ_286–290_ compared to full-length FliC, suggesting that the C-terminal region of the *C. difficile* flagellin is necessary for this phosphorylation. Whether the level of NLRC4 activation induced by FliC is lower than that induced by other flagellins remains to be determined.

The formation of the inflammasome involves the recruitment of pro-caspase-1. Using confocal microscopy, the colocalization of the *C. difficile* flagellin with pro-caspase-1 inside the cell was detected, suggesting an NLRC4 inflammasome assembly. The detection of the activation of pro-caspase-1 by cleavage into subunits p10 and p20 confirmed this inflammasome formation [37]. As previously described for other flagellins, releasing of mature IL-18 by Caco-2/TC7 cells following internalization of FliC, suggest a role for caspase-1 in the maturation of pro-IL-18. Downstream inflammasome activation pathway, the activated caspase-1 elicits the maturation (cleavage) of the GSDMD protein at its N-terminal domain, to form cell membrane pores leading to pyroptosis [38,39]. A structural study showed that the gasdermin pore is formed by 27 or 28 gasdermin monomers creating a 180 Å inner diameter channel allowing the transport of cytokines and numerous ions [40]. Decreasing the full-length GSDMD level induced by the *C. difficile* FliC strongly suggests a cleavage of this protein and its role in pyroptosis which could contribute to the inflammatory response during *C. difficile* infection.

Transcriptional regulation through NF-κB or interferon regulatory factor (IRF) affects most, if not all, inflammasome molecules from sensors to downstream inflammasome products such as caspase, pro-inflammatory cytokines, and GSDMD. For example, in human monocytes, pro-caspase-4 and pro-caspase-1 gene expressions are regulated by IRF2 [41,42]. Moreover, caspase-11 expression is regulated in the presence of lipopolysaccharide (LPS) or interferon gamma (IFNγ) via the NF-κB signaling pathway [43]. In this study, the observed FliC-induced upregulation of the caspase-1 as the inflammasome-related IL-1β IL-18 and IL-33 cytokine encoding genes was dramatically inhibited by chemical inhibition of NF-kB suggesting a role of NF-kB in the transcriptional expression of inflammasome products following FliC stimulation. In this way, as described for NLRP3 inflammasome [26], NF-κB could play an essential role in the activation of the inflammasome signaling pathway by preparing the pool of pro-caspase-1 and the not constitutively expressed pro-inflammatory inflammasome-related cytokines. Whether NLRP4-FliC inflammasome is similarly subjected to synergy from priming signals has yet to be determined.

Previous reports indicate that intracellular *C. difficile* toxins can activate the NLRP3 inflammasome and trigger IL-1bβ maturation which contributes to toxin-induced inflammation and intestinal injury [7]. Following the disruption of the epithelial barrier by altering tight junctions [27], the toxins interact with mucosal immune cells to elicit an intense inflammatory response leading to subsequent tissue damage [44,45]. However, the exact mechanism by which the toxin activates the inflammasome remains unknown. Moreover, the *C. difficile* surface layer proteins, forming a crystalline regular array that covers the surface of the bacterium [8], activates inflammasomes and pyroptosis via the ATP-P2X7 pathway which contributes to the inflammatory process and host defense [46].

In conclusion, the present study highlights the role of *C. difficile* flagellin in inflammation by activating the intracellular NLRC4 inflammasome signaling pathway in addition to its already-known role in the activation of the TLR5 pro-inflammatory pathway. In this sense, our results reinforce the idea of the role of *C. difficile* flagella in synergy with toxins in the inflammatory process responsible for lesions of the intestinal mucosa. In contact with epithelial cells, internalized *C. difficile* flagellin stimulates the inflammasome pathway, thus escaping the tolerance of the host’s immune system to bacterial flagellins present in the intestinal lumen. Therefore, FliC contributes with toxins and other surface structures to the induction of the inflammatory response of the host by stimulating receptors/sensors of the innate response both on the cell surface and intracellularly. Further studies are necessary to better characterize the internalization mechanisms of FliC inside the intestinal epithelial cells as well as the specific effectors involved in the *C. difficile* FliC stimulated NLRC4 signaling pathway in the context of CDI to better understand the pathogenesis of the bacterium. This study contributes to an improved understanding of *C. difficile* pathogenesis and host immune responses to *C. difficile* infection, which may facilitate the development of new host-targeted therapeutic and successful vaccine strategies.

## 4. Materials and Methods

Cell culture. Human intestinal epithelial Caco-2/TC7 cell line was grown in Dulbecco’s modified Eagle’s medium (DMEM; Gibco Laboratories, Grand Island, NY, USA) with L-glutamine (Gibco Laboratories) supplemented with 15% heat-inactivated fetal bovine serum (FBS; Gibco Laboratories) and 1% non-essential amino acids (Gibco Laboratories). Cells were maintained at 37 °C and 10% CO_2_.

Construction of recombinant pET28a-GFP-FliC and pET28a-GFP-FliCΔ_286–290_. The GFP gene and the FliC gene were amplified from the plasmids pCA24N-GFP and pET28a-FliC [12] respectively with specific primers (Table 1) and cloned in the pET28a vector using the Golden Gate method (New England BioLabs, Ipswich, MA, USA) [47]. The cloning resulted in the fusion of the GFP with the N-terminus of FliC. Briefly, a pET28a vector carrying *BsaI* restriction sites at each side of a spectinomycin gene was mixed with the two amplicons in the presence of *BsaI* and T4 ligase (New England Biolabs, Ipswich, MA, USA), for 1 h at 37 °C. Then *E. coli* TG1 was transformed with the ligation and after selection of the transformants on a kanamycin-containing Luria broth (LB) agar plate, plasmids were extracted and cloning verified by enzymatic digestion. The same procedure was used to create the pET28a-GFP-FliCΔ_286–290_ with FliC truncated of 15 nucleotides in 3′ by amplification with specific primers (Table 1). Each construct was further sequenced for verification.

Recombinant protein expression and purification. Recombinant proteins GFP-FliC and GFP-FliCΔ_286–290_ were expressed in the Escherichia coli BL21 strain transformed with the plasmids pET28a-GFP-FliC and pET28a-GFP-FliCΔ_286–290_ respectively, as described by Tran et al. with some modifications [48]. Each *E. coli* transformant was briefly grown in LB media supplemented with kanamycin at 37 °C for 18 h. The culture was then subcultured at 1:10 (*v*/*v*) and inoculated at 37 °C until OD600 reached 0.6. Induction was made in the presence of IPTG (1 mM concentration), and protein expression was performed at 16 °C for 18 h. Protein extraction was done by sonication on ice. 

Protein purification was performed by affinity chromatography using His-Select Cobalt Affinity Gel (Talon Superflow; Cytiva, Buckinghamshire, UK). After column equilibration with solution A (50 mM Tris-HCl, 100 mM NaCl, 20 mM imidazole pH 8.0), target protein was eluted with solution B (50 mM Tris-HCl, 100 mM NaCl, 200 mM imidazole, pH 8.0). Purified GFP-FliC (or GFP-FliCΔ_286–290_) samples were loaded on électrophorèse en gel de polyacrylamide contenant du dodécysulfate de sodium (SDS-PAGE, SDS 12%), and Coomassie Brilliant Blue stained gel confirmed the presence of purified proteins as the expected molecular weight (GFP-FliC: 69 kDa, GFP-FliCΔ_286–290_: 66 kDa).

Stimulation and treatment cell lines. For inflammasome activation experiments, cells were seeded on 24-well cell culture plates (5 × 10^5^ cells/well) and grown up to 3 days post-confluence. For immunofluorescence experiments, cells were seeded on 24-well cell culture plates (15x10^4^ cells/well) and grown up to 60–70% confluence. Cell monolayers were FBS depleted for 4 h before signaling experiments to reduce the phosphorylation background induced by the growth-factor-rich FBS. Cell stimulation with flagellins (recombinant FliC, recombinant GFP-FliC, or recombinant GFP-FliCΔ_286–290_) was assessed by incubation of cell monolayers with 5 µg/ml of recombinant protein. When indicated, positive controls were obtained by incubating cells with 1.25 µg/ml TNFα (InvivoGen, San Diego, CA, USA) for 15 min or 2 µg/ml ultrapure flagellin from S. Typhimurium (FlaST, InvivoGen; San Diego, CA, USA) for 2 h, 4 h, 6 h or 8 h, or 150 µg/ml LFn-Needle (InvivoGen; San Diego, CA, USA) for 2 h, or 5 µg/mL recombinant GFP for immunofluorescence quantification. Inhibition of pro-caspase-1 was obtained by adding 12.5, 25, or 50 µM VX-765 (Caspase-1 inhibitor, InvivoGen, San Diego, CA, USA) to cell monolayers 4 h before stimulation with recombinant proteins [49]. Inhibition of NF-κB was obtained by adding 20 µM BMS-345541 (IΚΚα inhibitor; Sigma-Aldrich, St. Louis, MO, USA) to cell monolayers 1 h before stimulation with recombinant proteins [50].

SDS-PAGE and Western blot analysis. Cells were washed with sterile PBS-Vanadate (0.5 mM EDTA, 1 mM sodium fluoride, 10 mM hydrogen peroxide, 0.1 mM sodium vanadate, PBS) at 37 °C to remove non-adhering particles, and frozen at −80 °C for at least 15 min. Cells were then lyzed by adding lysis buffer (186 mM β-Mercaptoethanol, 1% bromophenol blue, 10 mM NaF, 25 mM NaPPI, 1 mM Na_3_VO_4_). Proteins were separated by SDS-PAGE (SDS 10%) and transferred to a polyvinylidene difluoride (PVDF) membrane (ThermoScientific, Rockford, IL, USA). For immunoblotting, membranes were washed with TBS (1X Tris-buffered Saline) 0.1% Tween 20, blocked in TBS (0.1% Tween 20, 5% milk), and probed separately overnight with antibodies. According to the manufacturer’s instructions, blots were then incubated with horseradish peroxidase (HRP)-linked secondary antibodies (described below), followed by chemiluminescence detection with the ECL Plus kit (Millipore, Billerica, MA, USA). Chemiluminescence signals were detected with a Fusion FX Imaging System (Vilber Lourmat, Marne La Vallée, France), analyzed with Fusion-CAPT software, and normalized with actin.

Primary antibodies used include rabbit polyclonal anti-NLRC4 (NP541; 1:1000, ECM biosciences; Versailles, KY, USA), anti-Caspase-1 (D7F10; 1:1000, Cell Signaling Technology; Danvers, MA, USA), anti-p20 (D57A2; 1:1000, Cell Signaling Technology; Danvers, MA, USA), anti-GSDMDC1 (sc-81868; 1:1000, Santa Cruz Technology; Dallas, TX, USA) or anti-actin (A2066; 1:10000, Sigma-Aldrich; St. Louis, MO, USA). Secondary antibodies used include anti-Rabbit IgG (7074; 1:10000, Cell Signaling Technology; Danvers, MA, USA) and anti-Mouse IgG (7076; 1:10000, Cell Signaling Technology; Danvers, MA, USA). 

Cytokine assays. After cell stimulations at the indicated time, cell-free culture supernatants were collected and stored at −80 °C until cytokine assays. According to the manufacturer’s instructions, human pro-inflammatory cytokine IL-18 production by Caco-2/TC7 cells was quantified using an ELISA-based method using Quantibody Human Inflammation Array 1 from RayBiotech (Norcross, GA, USA).

Quantitative real-time reverse transcription PCR. Total RNA from cell lines was isolated using the RNeasy Mini Kit (Qiagen, Hilden, Germany). cDNA was prepared from 1 µg RNA using SuperScript™ III Reverse Transcriptase (Invitrogen, Carlsbad, CA, USA) with random primers as described by the manufacturer. qPCR was performed in a 10 µL reaction volume containing 4 ng of cDNA, 5 µL of SSo Advanced™ SYBR Green Supermix (Bio-Rad; Marnes-la-Coquette, France), and 500 nM gene-specific primers. The primers are listed in Table 2. Reactions were run on a CFX96 Real-time System (Bio-Rad; Marnes-la-Coquette, France) with the following cycling conditions: 30 s polymerase activation at 95 °C and 40 cycles at 95 °C for 5 s and 60 °C for 10 s. An additional step from a start at 65 °C to 95 °C (0.5 °C/0.5 s) was performed to establish a melting curve to verify the specificity of the real-time PCR reaction for each primer pair. Results were normalized using the geometric averaging of the reference gene *GAPDH*. Normalized relative ratios were calculated using the ∆∆CT method.

Immunofluorescence assays and quantification. Cells were plated in 24-well culture plates on glass coverslips. After 6 h of incubation with GFP-FliC, cells were washed with 1X PBS (pH 7.2), fixed with 3% paraformaldehyde, and permeabilized with 0.1% Triton in PBS. Cells were then washed three times with PBS (pH 7.2), blocked with PBS-gelatin 0.2%, and then incubated with primary antibody rabbit polyclonal caspase-1 (sc-56036, 1:100, Santa Cruz Biotechnology; Dallas, TX, USA) which detects C-terminus of full-length and cleaved caspase-1 for 3 h at RT. After washing with 1X PBS, proteins of interest were detected by incubation for 1 h with fluorescently labeled secondary antibodies AlexaFluor (Abberior Star Red; Sigma-Aldrich, St. Louis, MO, USA) 638 goat anti-rabbit IgG (kindly donated by V. Nicolas, PlateForme d’Imagerie Cellulaire—MIPSIT; 1:200), then washed with 1X PBS. Cells were mounted in ProlongTM Gold antifade reagent, containing DAPI staining the nuclear DNA (P36934, Invitrogen Carlsbad, CA, USA). F-actin staining was performed using Phalloidin–Tetramethylrhodamine B isothiocyanate 560 (P1951, Sigma-Aldrich, St. Louis, MO, USA) reagent according to the manufacturer’s instructions. Samples were observed with an inverted g-STED TCS SP8 Leica confocal microscope (Leica, Germany) using an HC PL APO CS2 63 ×/1.40 oil immersion objective lens.

For the FliC internalization analysis, taken images are composed of arithmetic stacks of 4–15 deconvoluted images, each 1.2 µm in z-step. Stacks of images (16-bit grayscale) were acquired with a z-step of 0.7 µm with low illumination intensity. GFP protein was used as a negative control for the quantification of the fluorescence intensity. The MFI) of GFP and GFP-FliC was determined by measurement of fluorescence in 30 representative fields (at least 100 individual cells), using Leica SP8 acquisition and software analysis (Imaris/ImageJ software). The histogram feature of internalization was used to compare MFI with the control (GFP). The two groups were compared using Mann and Whitney’s test performed with Graphpad software. *p* values < 0.05 were considered statistically significant.

Statistical analysis. Results are expressed as means ± sem. Different groups were compared with Mann and Whitney test or Student *t*-test performed by GraphPad Prism software. *p* values < 0.05 were considered statistically significant.

## Figures and Tables

**Figure 1 ijms-23-12366-f001:**
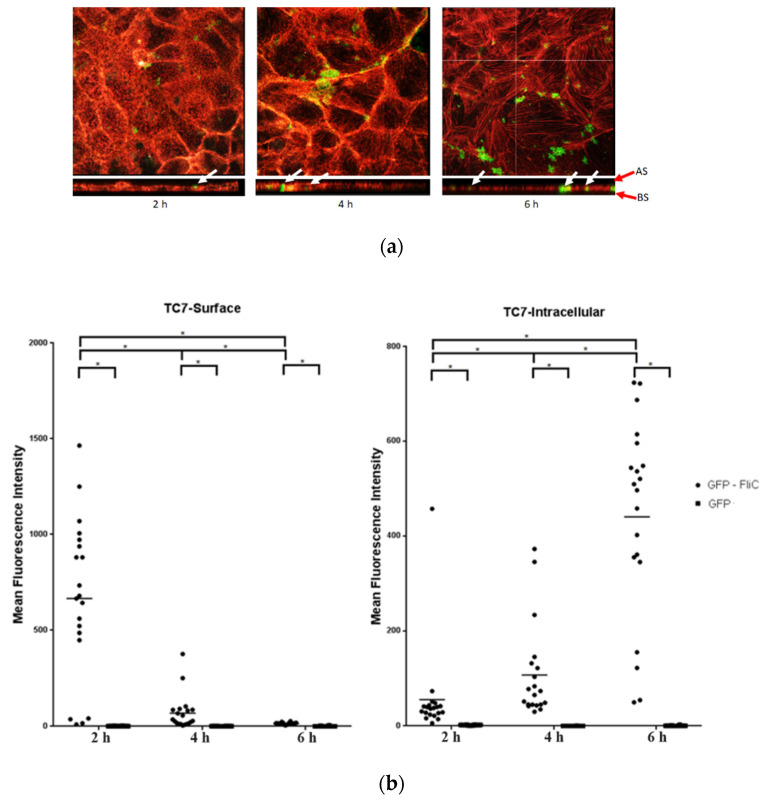
Internalization of the *C. difficile* flagellin in Caco-2/TC7 cells analysis by confocal microscopy. Caco-2/TC7 cells were incubated with the GFP-FliC (6 µg/mL) or GFP, as a control, for 2 h, 4 h, and 6 h. (**a**) Using the Imaris software, we analyzed the profile xz of cells and studied the internalization of GPF-FliC by the epithelial cells. Image acquisition of subsequent sections of transversal planes (from apical surface (AS) to basolateral surface (BS), red arrows) and longitudinal planes of the cells. White arrows point out the GFP-FliC proteins in longitudinal planes. (**b**) Mean Fluorescence Intensity (MFI) of GFP-FliC (●) and GFP (■) at the cell surface (TC7-Surface) and inside Caco-2/TC7 cells (TC7-Intracellular). Results represent MFI for 30 representative fields for each time. Horizontal lines represent the mean for each condition. * Statistically significant differences (*p* < 0.05).

**Figure 2 ijms-23-12366-f002:**
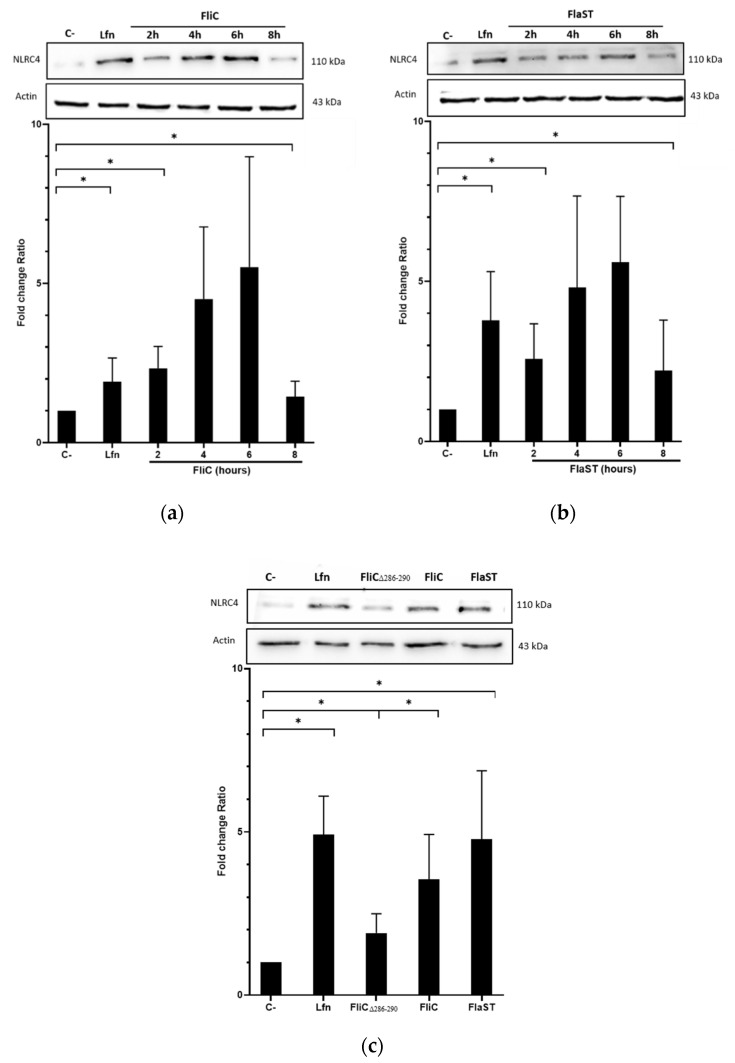
NLRC4 activation by *C. difficile* flagellin. Caco-2/TC7 cells were stimulated with (**a**) the *C. difficile* FliC (6 µg/mL) or (**b**) FlaST (2 µg/mL) for 2 h, 4 h, 6 h and 8 h, and a positive control LFn-Needle (150 µg/mL) for 2 h. (**c**) Cells were stimulated with the *C. difficile* FliC (6 µg/mL), FliCΔ_286–290_ (6 µg/mL), FlaST (2 µg/mL) for 4 h, and a positive control LFn-Needle (150 µg/mL) for 2 h. Western Blot was then performed using anti-phosphorylated (Ser533) NLRC4 and anti-actin antibodies. Images are representative of three independent experiments. The density of the bands was measured using Fusion software. Ratios NLRC4/actin were calculated and normalized to a negative control. Results represent the mean (*n* = 3) ± standard deviations for each condition. * Statistically significant differences (*p* < 0.05).

**Figure 3 ijms-23-12366-f003:**
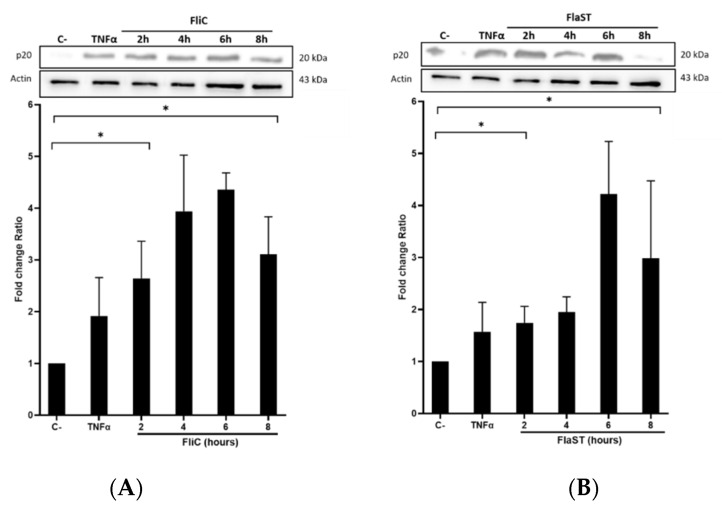
*C. difficile* flagellin induces caspase-1 and GSDMD activation. (**A**–**D**) Release of p20 after stimulation of cells with *C. difficile* flagellin. Caco-2/TC7 cells were stimulated with TNFα (1.25 µg/mL) for 1 h as a positive control and (**A**) *C. difficile* FliC (6 µg/mL) or (**B**) FlaST (2 µg/mL) for 2 h, 4 h, 6 h, and 8 h. Cells were pre-treated, or not, with caspase-1 inhibitor VX-756 (12.5, 25, or 50 µM) for 4 h and then stimulated with (**C**) *C. difficile* FliC (6 µg/mL) for 4 h, or (**D**) with LFn-Needle (150 µg/mL) for 2 h, as a positive control. (**E**–**F**) GSDMD cleavage by *C. difficile* flagellin. Cells were pretreated, or not, with caspase-1 inhibitor VX-756 (12.5, 25 and 50 µM) during 4 h and then stimulated with FliC (6 µg/mL) for 4 h, or a positive control LFn-Needle (150 µg/mL) for 2 h. After cell stimulations, western blots were then performed using anti-p20 or anti-GSDMD, and anti-actin antibodies. Images are representative of three independent experiments. The density of the bands was measured using Fusion software. Ratios of p20/actin or GSDMD/actin were calculated and normalized to a negative control. Results represent the mean (*n* = 3) ± standard deviations for each condition. * Statistically significant differences (*p* < 0.05). (**G**) Colocalization of *C. difficile* flagellin with pro-caspase-1. Cells were incubated with *C. difficile* GFP-FliC (6 µg/mL) for 6 h and assessed by immunofluorescence. Cells were labeled for (a) actin (magenta fluorescence excited with 560-nm light) and nuclei (blue fluorescence excited with 360-nm light), (b) FliC in green (green fluorescence excited with 488-nm light), (c) pro-caspase-1 labeled in red with AlexaFluor. (d) Merge between (b) and (c). (Scale bar 10 µm).

**Figure 4 ijms-23-12366-f004:**
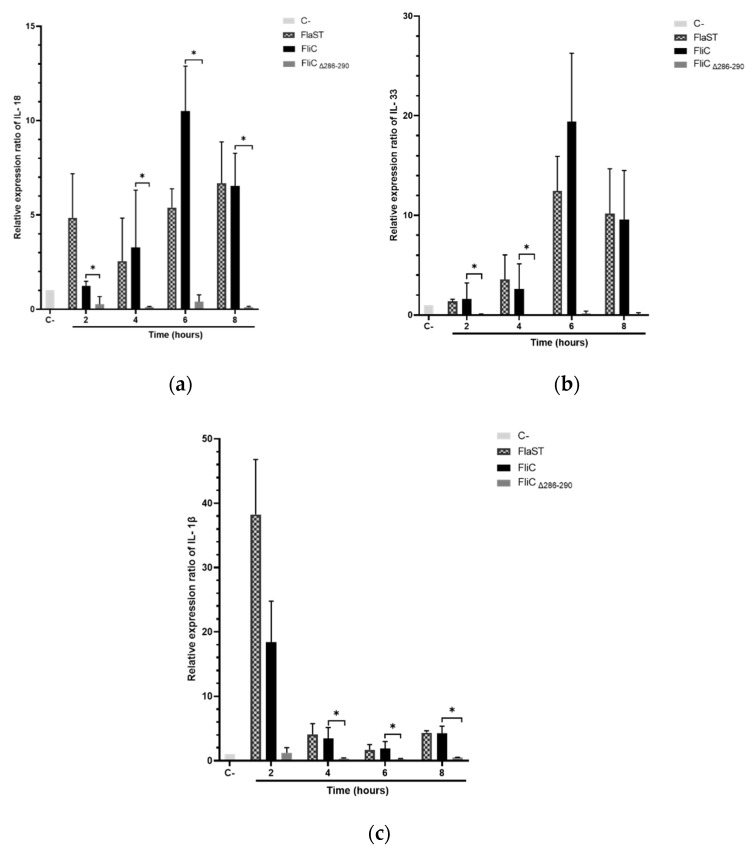
Inflammasome related-cytokines gene expression induced by *C. difficile* flagellin. Caco-2/TC7 cells were stimulated with FliC (6 µg/mL), FlaST (2 µg/mL) as a positive control, or FliCΔ_286–290_ (6 µg/mL) for 2 h, 4 h, 6 h, and 8 h, and monitored for *IL18* (**a**), *IL33* (**b**) and *IL1β* (**c**) gene expression by qRT-PCR. Results represent the mean (*n* = 3) ± standard deviations for each condition. * Statistically significant differences (*p* < 0.05).

**Figure 5 ijms-23-12366-f005:**
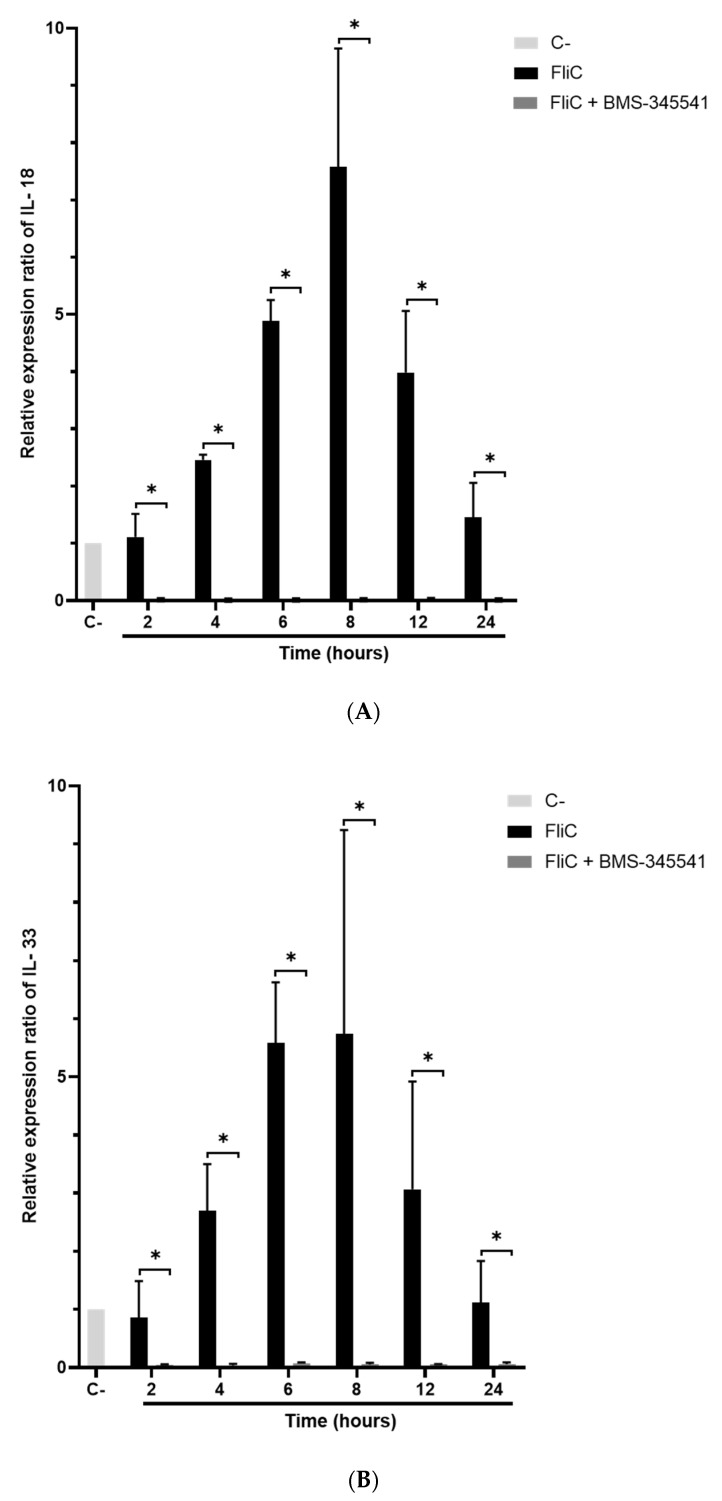
Abolition of FliC-induced pro-caspase-1 and inflammasome-related cytokines gene expression following inhibition of NF-κB signaling pathway. Caco-2/TC7 cells were pre-treated with the IΚΚα inhibitor BMS-345541 for 1 h, then stimulated with FliC in a time-dependent manner (2 h, 4 h, 6 h, 8 h, 12 h, and 24 h) and monitored for *IL18* (**A**), *IL33* (**B**) and *CASP1* (**C**) gene expression by qRT-PCR. Results represent the mean (*n* = 3) ± standard deviations for each condition. * Statistically significant differences (*p* < 0.05).

**Figure 6 ijms-23-12366-f006:**
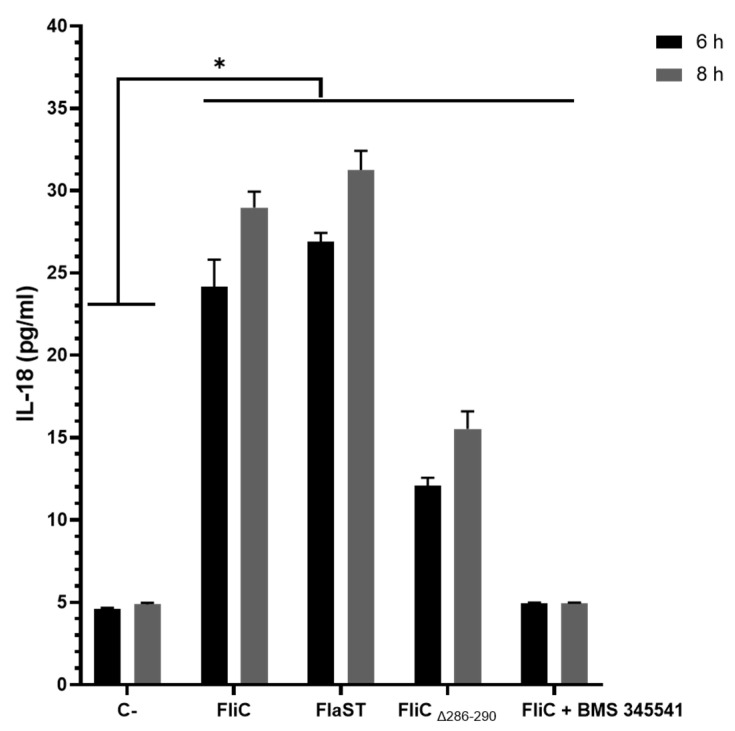
*C. difficile* flagellin induces IL-18 synthesis in Caco-2/TC7-cells. Cells were inhibited for 1 h with a chemical inhibitor of NF-κB signaling pathway, or not, then incubated with FliC, FlaST, as a positive control, and FliCΔ_286–290_, for 6 h, or 8 h, and IL-18 was quantified from supernatants. Errors bars indicate the standard error of the mean of three independent experiments. * Statistically significant differences (*p* < 0.05).

**Table 1 ijms-23-12366-t001:** Primers sequences for PCR.

Gene Name	Sequence
*GFP*	5′-C GGT CTC ATT GCT ATG CGG CCG CAG TAA AGG AG-3′ 5′-C GGT CTC ATT TGT ATA GTT CAT CCA TGC CAT G-3′
*fliC*	5′-C GGT CTC ACA AAA TGA GAG TTA ATA CAA ATG TAA GTG C-3′ 5′-C GGT CTC ATG CCG CTC CTA ATA ATT GTA AAA CTC C-3′
*fliC*Δ_286–290_	5′-C GGT CTC ACA AAA TGA GAG TTA ATA CAA ATG TAA GTG C-3′ 5′-C GGT CTC TTT GCA AAC TCC TTG TGG TTG TTG ATT AGC-3′

**Table 2 ijms-23-12366-t002:** Primers sequences for qRT-PCR.

Gene Name	Sequence
*hpro-caspase-1*	5′-GTT TCT TGG AGA CAT CCC ACA-3.’ 5′-TGG TGG GCA TCT GCG CTC TAC-3′
*hNLRC4*	5′- AAT GCA AAG AGG TCA TCG C-3.’ 5′-AGA GCC TTG CCA AGA GAA GA-3′
*hIL-8*	5′-GGC ACA AAC TTT CAG AGA CAG-3′ 5′-ACA CAG AGC TGC AGA AAT CAG G-3′
*hIL-18*	5′-AAA GAT GGC TGC TGA ACC AGT-3′ 5′-TTT CCT CAG CTG ACA ATG GTG-3′
*hIL-33*	5′-AGA ACT GGG ATG TAA CTG CCT-3′ 5′-CTT TGC TTG CTG TGT TCT TCC-3′
*hIL-1β*	5′-AAT TTG AGT CTG CCC AGT TCC CC-3′ 5′-AGT CAG TTA TAT CCT GGC CGC C-3′
*hGAPDH*	5′-AGC CTT CTC CAT GGT GGT GAA GAC-3′ 5′-CGG AGT CAA CGG ATT TGG TCG -3′

## Data Availability

Not applicable.

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
