# Peer review of "Clostridioides difficile Flagellin Activates the Intracellular NLRC4 Inflammasome"

_ijms, 2022, doi:10.3390/ijms232012366_

Round 1

Reviewer 1 Report

Reviewer Reports:

I recommend a major amendment at this level.

General comments:

The manuscript entitled Clostridioides difficile flagellin activates the intracellular NLRC4 inflammasome” was reviewed. The work carried out in the manuscript is interesting and aimed to assess the internalization of C. difficile FliC inside the in testinal epithelial cells by confocal microscopy. The manuscript has a lot of information however, there is some correction needed before possible publication. It is better to do not to use the first person's pronoun. Do not use "we, us, or our" throughout the paper. The authors are suggested to proofread the paper with a native English speaker and restructuring of sentences is required for the entire manuscript. What progress against the most recent state-of-the-art similar studies was made? The innovation and the importance of this work are not clearly highlighted in the abstract, introduction and conclusions. Please work on this and prove to us why this work is valuable. The structure of this work should be reorganized. For example, the Section on the method should be after the introduction. Please also remove ANY lumped references. Please define each of them separately to avoid inappropriate citations. Other main remarks that in my opinion needs attention are the following:

Detailed comments:

Title:

Ok.

Abstract:

The abstract does work well. However, a good abstract should address these issues: what are you trying to do, why, what you found and what is the significance of your findings. In the abstract, please add an indication of the achievements from your study that are relevant to the journal scope. The abstract should state briefly the purpose of the research, the principal results and major conclusions. In the abstract, please add an indication of the achievements from your study that are relevant to the journal scope. Please be concise - maximum 1-2 lines.

Introduction:

The literature review should clarify the "contribution" of your study. The authors failed to present the study debates and failed to discuss the debates. In general, the authors should present the specific debate for your study. This should more clearly show the knowledge gaps identified and link them to the paper goals. A high-quality paper has to provide a proper state-of-the-art analysis after the literature review and only based on the analysis to formulate the paper's goals. The lack of proper justification creates the wrong impression that the authors are unaware of the recent developments.

Please eliminate the use of redundant words. Eg. In this way, Recently, Respectively, therefore, currently, thus, hence, finally, to do this, first, in order, however, moreover, nowadays, today, consequently, in addition, additionally, furthermore. Please revise all similar cases, as removing these term(s) would not significantly affect the meaning of the sentence. This will keep the manuscript as CONCISE as possible. Please check ALL. Avoid beginning or ending a sentence with one or a few words, they are usually redundant. Kindly revise all.

Materials and Methods:

The materials and methods have been written in sequence.  Please support and quote more references and update with recent references. please add in the beginning your scientific hypothesis. In the course of describing the performed actions, please provide reader guidance, sufficient for understanding why those actions have been performed. Please avoid having one heading after another with no discussion in between as in the case of Sections 4 and 4.1. Kindly inspect the entire document for similar instances and revise accordingly. The percentage purity and company of all reagents/chemicals utilized must be reported. Though some of the model/brands of the equipment used were stated, their country of manufacture should be reported as well.

Results and Discussion:

The structure of this work should be reorganized. For example, the Section of results should be combined with the Discussion. All the findings of the current work need to be compared and discussed with the results of other researchers finding instead of having a general comparison with other researchers' works. The authors should perform a comparison between the forecasting results. In your discussion section, please link your empirical results with a broader and deeper literature review.

Conclusions:

Missing????

References:

Please check the reference section carefully and correct the inconsistency.

Reviewer 2 Report

The manuscript needs better organisation. The discussion and introduction sections need more information with additional supporting references. The total number of references is very low and does not adequately cite relevant literature. The conclusion is completely missing.  Overall, the manuscript needs significant improvement before it can be acceptable for publication in your journal.

Round 2

Reviewer 1 Report

Reviewer Reports:

I have reviewed the revised version manuscript entitled” Clostridioides difficile flagellin activates the intracellular NLRC4 inflammasome”. The paper has been improved and can be accepted. 

Reviewer 2 Report

The authors have addressed the comments.